# Splitting with Importance-aware Updating for Heterogeneous Federated Learning with Large Language Models

Yangxu Liao [* 1 2]  Wenke Huang [* 1 2]  Guancheng Wan [* 2]  Jian Liang [1 2]  Bin Yang [2]  Mang Ye [+ 2]

## Abstract

Federated learning provides an efficient privacy-preserving distributed training framework for large language models, addressing the growing scarcity of publicly available training data while enabling the utilization of private datasets. While integrating large language model fine-tuning with federated learning emerges as a promising research direction, researchers pay limited attention to non-IID instruction-following scenarios. Our key insight is decomposing client updates into consensus and divergence components, enabling the model to maintain core capabilities while adapting to domain-specific knowledge. We propose a novel federated learning framework called **FedICU** (Splitting with Importan**C**e-aware **U**pdating for Heterogeneous **Fed**erated Learning with Large Language Models), which introduces an aggregation mechanism that dynamically balances these components based on their contribution to global model performance, while implementing an importance-aware parameter updating strategy to prevent catastrophic forgetting and domain overfitting. Extensive experiments across diverse domains demonstrate that FedICU significantly outperforms existing federated learning approaches in terms of both generalization performance and domain adaptation. Our code is available at https://github.com/liaosunny123/FedICU.

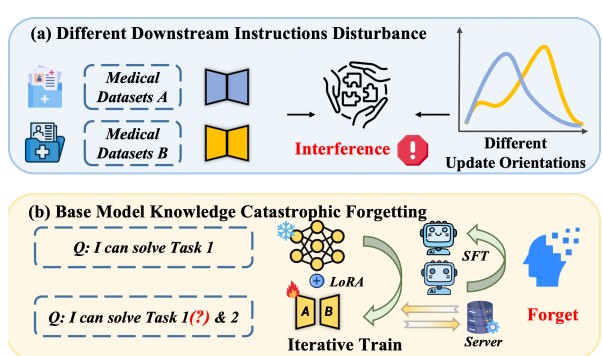

Figure 1: **Motivation**. In the current process of Large Language Model Supervised Fine-tuning, the following issues exist: (a) *Heterogeneous client distributions and varying instruction quality* in Supervised Fine-tuning impair global model performance. (b) *Ineffective preservation of general capabilities* during federated training of large language models fine-tuning.

## 1. Introduction

Federated learning is a collaborative paradigm, allowing multiple clients to train a shared global model collaboratively without privacy leakage (McMahan et al., 2017b; Reddi et al., 2020; McMahan et al., 2017a; Huang et al., 2022; 2023c). Large language models (LLMs) (Achiam et al., 2023; Ouyang et al., 2022; Touvron et al., 2023; Devine, 2024; Liu et al., 2023; Wang et al., 2024) emerge as a transformative technology across main fields in recent years (Imani et al., 2023; Didolkar et al., 2024; Chen et al., 2023; Fang et al., 2025a; Jiang et al., 2025). While trained on large public datasets, these large language models demonstrate significant success in solving general problems across various domains. However, the scarcity of high-quality public training data becomes a critical bottleneck for large language model development (Kaddour et al., 2023), with predictions suggesting the exhaustion of such data by 2026 (Villalobos et al., 2022). Although existing approaches attempt to address this challenge through datasets combination (Wang et al., 2023) or self-generated data (Wang et al., 2022), these methods often fall short - the former is limited by data availability (Kaplan et al., 2020), while the latter risks model degradation (Alemohammad et al., 2023;

---

[*]Equal contribution [1]Guangdong Laboratory of Artificial Intelligence and Digital Economy (SZ), Shenzhen, China [2]National Engineering Research Center for Multimedia Software, School of Computer Science, Wuhan University, Wuhan, China. Correspondence to: Mang Ye <yemang@whu.edu.cn>.

*Proceedings of the 42nd International Conference on Machine Learning*, Vancouver, Canada. PMLR 267, 2025. Copyright 2025 by the author(s).

Muennighoff et al., 2023). Although substantial private datasets exist, their direct utilization is frequently restricted by privacy concerns. Notable large language models, such as BloombergGPT (Wu et al., 2023), FinGPT (Yang et al., 2023b) and etc., successfully leverage private datasets, highlighting the potential value of these resources. This situation presents an opportunity for federated learning, which enables the utilization of private, high-quality datasets while maintaining privacy (Li et al., 2020), potentially offering a solution to the ongoing challenge of data scarcity in large language model development (Zhuang et al., 2023).

To address these challenges, numerous researchers develop diverse frameworks to train or fine-tune large language models on geographically distributed private datasets (Li et al., 2024b; Fan et al., 2023; Xu et al., 2023b). In particular, researchers focus on integrating the large language model Supervised Fine-Tuning (SFT) module (Gunel et al., 2020; Ye et al., 2024) into the federated learning framework, such as FederatedScope-LLM (Kuang et al., 2024), Shepherd (He et al., 2021) and OpenFedLLM (Ye et al., 2024). These studies utilize LoRA for fine-tuning the global model on geographically distributed datasets, enhancing the model capabilities while allowing it to leverage knowledge from private datasets. However, these frameworks demonstrate an excessive focus on classical federated learning algorithms, simply embedding the LLM supervised fine-tuning process within the federated learning framework without considering the unique characteristics of language model training, such as the base ability and the domain-specific knowledge. This oversight potentially limits the effectiveness of federated learning in scenarios where LLMs need to maintain their general capabilities while acquiring domain-specific knowledge. In a nutshell, the aforementioned discussions motivate us to rethink: *How can we effectively incorporate intrinsic characteristics of the LLM, particularly the interaction between general and domain-specific knowledge, into federated fine-tuning frameworks?*

Preliminarily, numerous existing works perform LLM supervised fine-tuning through FedAvg. In this paradigm, each local dataset operates an independent client that receives the LoRA parameters distributed from the global LLM in the server. These clients conduct training on their local datasets and transmit the updated gradients back to the global model. The global model then averages these gradient updates and iteratively repeats this process. However, when dealing with heterogeneous downstream instructions of different clients, simple averaging of client-side LoRA parameters may not achieve optimal performance, since the global LLM typically exhibits bias toward specific local distributions rather than maintaining true global optimality. Considering that LLM training typically involves pre-training a foundation model followed by fine-tuning, we hypothesize the existence of two distinct factors: one that maximally preserves the

foundation model capabilities, and another that optimally captures domain-specific characteristics. To investigate this hypothesis, we perform decomposition analysis on LoRA parameters and discover **convergence factor**, which express the general ability of the model, and **divergence factor**, which expresses the especial ability of the model, in the process of supervised fine-tuning of LLM on the federated learning. Based on this observation, we raise the following question: ***1) How can we ensure effective instruction learning during federated fine-tuning on the heterogeneous instruction preferences and varying quality levels?*** Furthermore, in terms of client-side updates, we find that extracting local features during client training is crucial for the global model. An effective training method should better capture features from datasets, perform sparse updates to the global model to preserve the base model ability and improve communication efficiency. However, in the current method, the state-of-the-art model algorithm found by Ye et al. (2024) simply uploads all the LoRA parameters. This indiscriminate parameter aggregation can result in the global model overfitting to specific client distributions while losing its general capabilities. Based on this, we further propose another question: ***2) How can we preserve and transfer the general capabilities of LLMs while enabling them to adapt to domain-specific knowledge?***

To address the two issues mentioned above, this paper proposes an innovative framework to help the global model to absorb the instruction knowledge from heterogeneous clients, enhancing generalization. To address problem 1, we propose the Consensus-Divergence Splitting, which combines consensus aggregation and divergence alignment, optimizing the global aggregation process and improving the performance of the global model. For problem 2, during local updates, we introduce the Importance-Aware Updating, which focuses on uploading significantly altered parameters while disregarding minimal changes. The mask enables the global model to accurately capture the direction of meaningful updates, addressing the problem of catastrophic forgetting and improving communication efficiency.

Our primary contribution in this paper can be summarized as following three points:

(1) We discover that LLM fine-tuning can be decoupled into consensus factor and divergence factor, and identify their specific meaning in the SFT, which has practical significance in the federated large language model fine-tuning.

(2) We propose a novel framework for federated fine-tuning of large language models during the SFT stage. To address domain knowledge drift across clients, we adopt a hybrid optimization strategy that combines consensus-based aggregation with divergence alignment. This ensures balanced integration of knowledge from diverse downstream tasks. During the client uploading process, we focus on making pa-

rameter updates effective and meaningful. Applying sparse updates to the global model, we preserve previously learned knowledge and maintain the model's generalization ability.

(3) Through extensive experiments, we demonstrate that our framework achieves significant improvements in model generalization ability compared to existing approaches.

## 2. Related Work

### 2.1. Parameter Efficient Fine-Tuning for Large Language Models

The emergence of Large Language Models (LLMs) has demonstrated remarkable capabilities across various natural language processing tasks (Achiam et al., 2023; Touvron et al., 2023; Dubey et al., 2024; Li et al., 2024a; Wang et al., 2024; Huang et al., 2023b; Bi et al., 2025a;c). However, adapting these models to downstream tasks through conventional fine-tuning poses significant computational and resource challenges (Houlsby et al., 2019; Valipour et al., 2022; Bi et al., 2025b; Fang et al., 2025b; 2024). In response to these challenges, the development of Parameter Efficient Fine-Tuning (PEFT) methods has gained significant attention, as these methods aim to reduce computational overhead while maintaining model performance (Xu et al., 2023a). Current approaches can be categorized into three main directions: **i)** *Adapter-based methods* (Houlsby et al., 2019; Pfeiffer et al., 2020; He et al., 2021; Edalati et al., 2022) introduce additional neural network layers for task-specific adaptation. These methods insert trainable modules between the original transformer layers while keeping the pre-trained parameters frozen, thereby enabling efficient parameter-sharing across tasks while maintaining model performance. **ii)** *Prompting techniques* (Petrov et al., 2023; Li & Liang, 2021) modify input sequences to guide model behavior through learnable or hard-coded prompts, effectively allowing models to leverage task-specific knowledge without extensive fine-tuning of the base model parameters. **iii)** *Low-rank adaptation* approaches, particularly LoRA (Hu et al., 2021), decompose weight updates into low-rank matrices ($\theta' = \theta + BA$), significantly reducing the number of trainable parameters while preserving the model ability to learn task-specific adaptations. This approach is founded on the observation that the weight updates during fine-tuning often have a low intrinsic rank. While LoRA has become the dominant approach in SFT due to its efficiency, its effectiveness in distributed settings remains largely unexplored.

### 2.2. Federated Learning for Large Language Models

Federated learning has emerged as a promising paradigm for privacy-preserving collaborative training (Kairouz et al., 2019; Wan et al., 2024; 2025; Yang et al., 2023a; Huang et al., 2023a; Wan et al., 2024). Its integration with large language models effectively addresses the critical challenge of leveraging private datasets while maintaining data privacy (Li et al., 2024b). Several recent works have made initial attempts in this direction. Fan et al. (2023) explored traditional federated learning tasks for large language models fine-tuning, while Xu et al. (2023b) focused on improving computational efficiency on client devices. Additionally Kuang et al. (2024) investigated federated instruction tuning using FedAvg, and Ye et al. (2024) proposed a framework encompassing both SFT and RLHF processes, combining large language models training processes with classical federated learning algorithms using LoRA-based fine-tuning. However, these existing methods exhibit several critical limitations: **i)** Some primarily focus on computational efficiency without addressing different downstream instructions heterogeneity in federated settings. **ii)** The interaction between PEFT methods and federated learning remains poorly understood, especially in terms of parameter aggregation and convergence properties. To address these limitations, we propose a novel framework that specifically addresses these challenges by jointly considering the client upload process and global aggregation process in non-IID settings. Through our careful design of the consensus-divergence mechanism and importance-aware parameter masking update strategy, our method not only achieves robust performance but also maintaining the efficiency benefits of LoRA-based fine-tuning.

## 3. Methodology

### 3.1. Preliminaries

LoRA (Hu et al., 2021) is employed to enhance the efficiency of fine-tuning by focusing on the internal rank variations that occur during parameter updates in the fine-tuning process. For fine-tuning a pre-trained model $\theta \in \mathbb{R}^{d \times k}$, LoRA keeps the pre-trained model parameter matrix frozen and utilizes two lower-rank matrices, $\theta_A \in \mathbb{R}^{d \times r}$ and $\theta_B \in \mathbb{R}^{r \times k}$, to represent the update $\Delta\theta$. This process can be formulated as Equation (1):

$$\theta' = \theta + \underbrace{(\theta_A \times \theta_B)}_{\Delta\theta}, \tag{1}$$

where $r \ll \min(d, k)$ is the rank of the decomposition.

During training, $\theta$ remains frozen, while $\theta_A$ and $\theta_B$ are updated. Initially, $\theta_B$ is initialized using the Kaiming distribution, and $\theta_A$ is set to a zero matrix, ensuring $\theta' = \theta$ at the start. Any low-dimensional decomposition method can be applied for $\Delta\theta$, as shown in Dettmers et al. (2023) and Rajabzadeh et al. (2024).

In federated learning, multiple clients collaborate with a central server to train a global model on different datasets. In this framework, each client maintains its private dataset and performs local training, while the server coordinates the

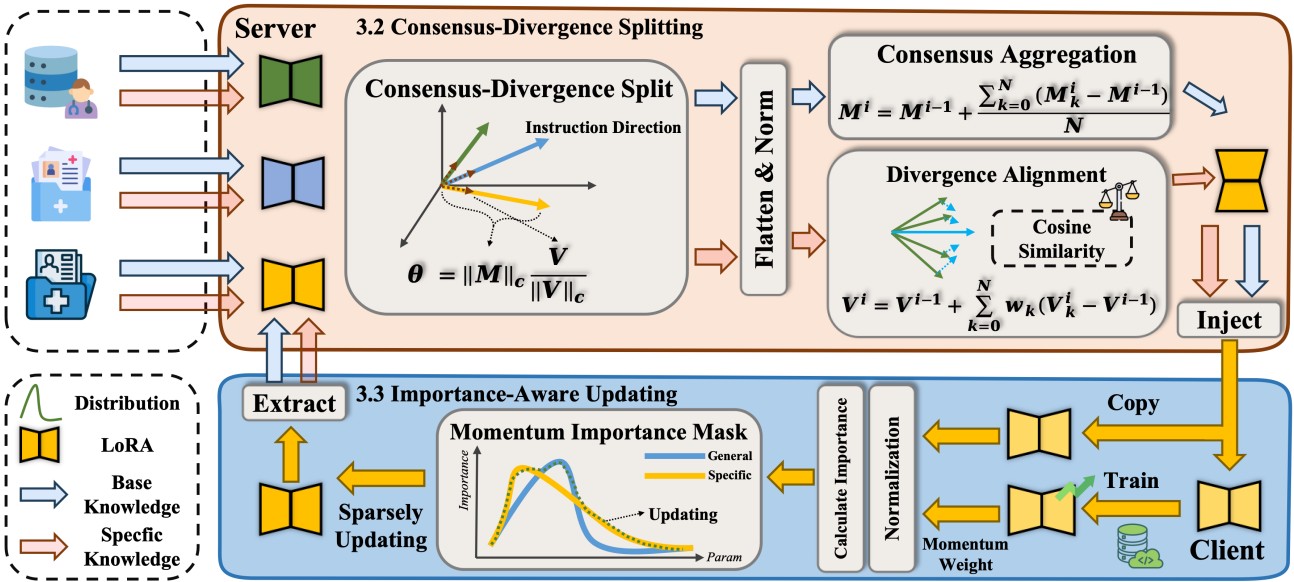

Figure 2: **Architecture illustration** of the Splitting and Updating components. The two key components are shown at the top (a) and bottom (b) of the image, with nodes of different classes marked in different colors. (a) Consensus-Divergence Splitting (Section 3.2) module splits the LoRA updates into consensus and divergence parts, applying consensus aggregation through delta averaging and divergence alignment using cosine similarity. (b) Importance-Aware Updating (Section 3.3) component computes every parameter importance based on the difference between the global model and the local model, selecting important knowledge from LLM clients for updates. Best viewed in color. Zoom in for details.

learning process through model aggregation. Specifically, during the $i$-th iteration, each client $k$ trains its local model $\theta_k^i$ using its own dataset and sends the model updates to the server. The server then aggregates these local models to generate an updated global model $\theta^i$, which is subsequently distributed back to the clients for the next round of training, where client $k$ obtains its next local model $\theta_k^{i+1}$ as the formula below Equation (2):

$$\theta_k^{i+1} \leftarrow \theta^i \leftarrow \sum_{k=1}^{N} \frac{n_k}{n} \theta_k^i, \qquad (2)$$

where $\theta_k$ is the model of $k$-th client, $n_k$ is the size of the dataset for the $k$-th client, $n$ is the total dataset size, and $N$ is the number of clients.

In Fed LLM training, a common approach to reduce communication costs is to utilize LoRA, which updates only a subset of parameters instead of the entire model. In the FedAvg method with LoRA, the global model $\theta^i$ is updated as follows Equation (3):

$$\Delta\theta^i \leftarrow \sum_{k=1}^{N} \frac{n_k}{n} \Delta\theta_k^i, \qquad (3)$$

where $\Delta\theta^i$ is the LoRA model of the global model at $i$-th round, and $\Delta\theta_k^i$ represents the LoRA model of the $k$-th local client model at $i$-th round.

We can compute the global model by adding the $\theta_A \times \theta_B$ to the frozen base model $\theta$, meaning that the global model $\theta'$ after training is given by Equation (4):

$$\theta' = \theta + \Delta\theta = \theta + \theta_A \times \theta_B, \qquad (4)$$

where $\theta_A \times \theta_B$ represents the LoRA matrix split by $\Delta\theta$.

However, two significant problems can arise with this training method: (1) Naive parameter aggregation across clients can impair model convergence and degrade general capabilities, especially when dealing with heterogeneous downstream instruction fine-tuning tasks (Guo et al., 2024; Zhu et al., 2021; Cho et al., 2024). (2) Not all knowledge acquired by clients from their private datasets contributes meaningfully to global model performance. Incorporating non-essential parameter updates can bias the global model toward suboptimal local minima. Moreover, indiscriminate parameter updates risk catastrophic forgetting, where the model loses previously acquired knowledge and suffers degraded generalization capabilities (Dou et al., 2024; Yang et al., 2024; Han et al., 2024; Zhu et al., 2024; Lee et al., 2022; Kim et al., 2024).

### 3.2. Consensus-Divergence Splitting

**Motivation.** Existing federated learning approaches for large language model supervised fine-tuning focus solely on aggregating client-side LoRA parameter matrices, failing to leverage interpretable nature of the LoRA or address the model drift that occurs when clients train on diverse instruction tasks (Zhu et al., 2021).

**Consensus-Divergence Splitting.** To address the challenge of heterogeneous clients training on different downstream instruction tasks in federated LLM supervised fine-tuning, we

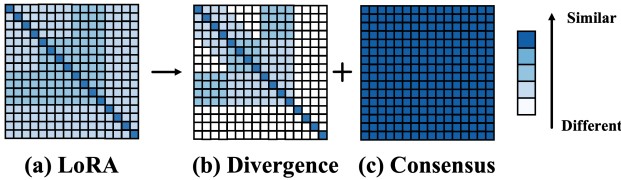

**(a) LoRA**    **(b) Divergence**    **(c) Consensus**

Figure 3: Visualization of similarity matrices between different updating parameters of all the clients different layers are shown as follows: (a) Similarity of LoRA matrices without decomposition, (b) Similarity of divergence matrices, and (c) Similarity of consensus matrices. The The results indicate that *the differences between distributed LLM behavior become more pronounced after decomposition into consensus and divergence.*

propose decomposing each client parameter matrix into magnitude and directional components inspired by Salimans & Kingma (2016). Our decomposition is motivated by two key observations: (1) the magnitude of parameter updates often reflects general instruction capabilities across clients, while (2) the directional changes capture client-specific adaptations, which are specialized for specific downstream tasks. Formally, for the LoRA update matrix $\theta^i \in \mathbb{R}^{r \times d}$ of client $k$ at $i$-th iteration, we split it using formula below, as Equation (5):

$$\Delta\theta_k^i = \underbrace{||\Delta\theta_k^i||_c}_{\text{consensus}} \cdot \underbrace{\frac{\Delta\theta_k^i}{||\Delta\theta_k^i||_c}}_{\text{divergence}}, \qquad (5)$$

where $||\Delta\theta_k^i||_c \in \mathbb{R}^r$ denotes the column-wise L2 norm that captures the scale of updates across the low-rank dimension $r$, and $\frac{\Delta\theta_k^i}{||\Delta\theta_k^i||_c}$ represents the normalized directions.

To empirically validate our approach, we conduct experiments with four clients on non-IID datasets and perform aggregation using the FedAvg (McMahan et al., 2017b). We compute the similarity of the LoRA matrices across different clients and visualize the results using a heat map, as shown in Figure 3.

Our analysis reveals that the client drift becomes more apparent after splitting parameter matrices into magnitude and direction components. Specifically, the differences between the clients are more pronounced in the Divergence Similarity Matrix, while the Consensus Similarity Matrix shows minimal variation between clients.

Based on these observations above, we can draw two key conclusions: (1) The decomposition method provides better discrimination of client differences compared to the base LoRA method. (2) The decomposition effectively separates two distinct aspects of the model: the direction matrix captures **client-specific divergence characteristics**, while the magnitude component reflects **shared consensus patterns** across heterogeneous clients.

**Consensus Aggregation.** Let $M_k^i = ||\Delta\theta_k^i||_c$ denote the

consensus vector of client $k$ at iteration $i$, which captures the magnitude component of the decomposed LoRA parameters and $M^i$ denote the global model consensus vector at iteration $i$. To aggregate these consensus vectors across clients, we compute the average change in updates relative to the previous global consensus state.

During local training, each client LLM extracts knowledge from its private dataset. We propose to aggregate these updates by averaging the differences relative to the previous global consensus, which ensures that new knowledge acquired by each client is incorporated uniformly into the global model, following the formula outlined below:

$$M^i = M^{i-1} + \frac{\sum_{k=0}^N (M_k^i - M^{i-1})}{N}. \qquad (6)$$

We can prove that this averaging strategy is optimal under certain conditions. Consider the consensus vector $M_k^i$ of the $k$-th client at iteration $i$, which can be modeled as the formula below, Equation (7):

$$M_k^i = M^* + \xi_k, \qquad (7)$$

where $M^*$ represents the true global consensus and $\xi_k \sim \mathcal{N}(0, \sigma^2 I)$ denotes the noise term. For any linear unbiased estimator $\hat{M} = \sum_{k=1}^N \alpha_k M_k^i$ satisfying $\sum_{k=1}^N \alpha_k = 1$, its variance is given by Equation (8):

$$\text{Var}(\hat{M}) = \sum_{k=1}^N \alpha_k^2 \sigma^2. \qquad (8)$$

Using the method of Lagrange multipliers, we can prove that the variance is minimized when $\alpha_k = \frac{1}{N}$ for all $k$ (see Appendix H). This theoretical foundation supports our practical approach of averaging the consensus vectors across clients to achieve optimal aggregation when the noise of the consensus is the same, which can be observed in Figure 3 and proved in a experiment conducted in the Appendix F.

**Divergence Aggregation.** Let $V_k^i = \frac{\Delta\theta_k^i}{||\Delta\theta_k^i||_c}$ denote the divergence vector of client $k$ at $i$-th iteration, which captures the divergence component of the decomposed LoRA parameters and $V^i$ denote the global model divergence vector at $i$-th iteration. To aggregate the divergence, we introduce an adaptive weighting scheme based on the pairwise cosine similarities between client updates.

For each client divergence vector $V_k^i$, we decompose it with LoRA down projection $A_k^i \in \mathbb{R}^{d \times r}$ and LoRA up projection $B_k^i \in \mathbb{R}^{r \times k}$. We perform vector-wise aggregation along their respective dimensions. Specifically, for the down projection matrix $A_k^i$, we iterate over columns to obtain $r$-dimensional vectors, while for the up projection matrix $B_k^i$, we iterate over rows to obtain $d$-dimensional vectors. For each vector slice, we compute the pairwise cosine similarities to form a similarity matrix $S \in \mathbb{R}^{n \times n}$, following the formula as Equation (9):

$$S_{pq} = \cos(v_p, v_q) = \frac{v_p \cdot v_q}{||v_p|| ||v_q||}, \qquad (9)$$

where $p, q$ index the clients, and $v_p$, $v_q$ represent the corresponding vector slices from clients $p$ and $q$.

We then compute each client $k$ average similarity score across all other clients as Equation (10):

$$s_k = \frac{1}{n-1} \sum_{j \neq k} S_{kj}, \qquad (10)$$

where $j \in \{1, 2, ..., n\}$.

The client contribution weights are determined through temperature-scaled softmax as Equation (11):

$$w_k = \frac{\exp(s_k/\tau)}{\sum_{j=1}^{n} \exp(s_j/\tau)}. \qquad (11)$$

Given the computed weights, we update the global divergence vector through a weighted aggregation of client updates relative to the previous global state as Equation (12):

$$V^i = V^{i-1} + \sum_{k=1}^{n} w_k(V_k^{i-1} - V^{i-1}), \qquad (12)$$

where $V^i$ and $V^{i-1}$ denote the global divergence vectors at iterations $i$ and $i-1$ respectively, $V_k^i$ represents the divergence vector of client $k$ at iteration $i$, $w_k$ is the computed weight for client $k$, and $n$ is the total number of clients.

Our weighting scheme ensures effective instruction learning during federated fine-tuning while accounting for client heterogeneity. The approach balances consensus compatibility with client-specific update directions through the divergence weights, effectively mitigating knowledge drift while preserving diverse instruction knowledge across clients.

**Rationale for Weighting** Our weighting scheme combines cosine similarity with temperature-scaled softmax, which can be theoretically formulated as the following optimization problem Equation (13):

$$\min_w \text{KL}(w|p) + T \sum_{k=1}^{n} w_k \log \frac{1}{\bar{s}_k}, \qquad (13)$$

where $\text{KL}(w|p)$ represents the KL divergence between weight distribution $w$ and uniform distribution $p$, and $\bar{s}_k$ denotes the average similarity score for client $k$.

The first term encourages weight diversity, while the second term enforces similarity consistency, with temperature parameter $T$ controlling the trade-off between these objectives. This optimization formulation naturally leads to our softmax weighting scheme in Equation (11). We choose cosine similarity as our metric due to its scale-invariance and computational efficiency in high-dimensional spaces, making it particularly effective for divergence vectors in large language models.

**Convergence Analysis.** We provide theoretical guarantees for the convergence of our proposed algorithm. The details can be found in the Appendix G.

---

**Algorithm 1** FedICU

**Input:** Communication rounds $T$, number of clients $N$, local training epochs $E$, temperature $\tau$, learning rate $\eta$, momentum coefficient $\beta$

**Output:** Final global LoRA model parameters $\theta^T$

Initialize global LoRA model parameters $\theta^0$ and client $k$ local model $\theta_k^0 = \theta^0$

Initialize momentum buffers $m_k^0 = 0$ for each client $k$

**for** $i = 1$ **to** $T$ **do**
    **Client-side**
    **for** each client $k \in \{1, \ldots, N\}$ **in parallel do**
        $\theta_k^i \leftarrow \theta^{i-1}$
        **for** local epoch $e = 1$ **to** $E$ **do**
            Update $\theta_k^i$ using local data
        **end**
        $\Delta\theta_k^i \leftarrow \theta_k^i - \theta^{i-1}$ // Equation (15)
        $m_k^i \leftarrow \beta m_k^{i-1} + (1-\beta)\Delta\theta_k^i$
        $I \leftarrow \sigma\left(\frac{|\theta^{i-1}| - \mu(|\theta^{i-1}|)}{\sigma(|\theta^{i-1}|) + \epsilon}\right)$ // Equation (14)
        $G \leftarrow \sigma\left(\frac{|m_k^i| - \mu(|m_k^{i-1}|)}{\sigma(|m_k^{i-1}|) + \epsilon}\right)$ // Equation (16)
        $W[v] \leftarrow \begin{cases} 1 & \text{if } G[v] > I[v] \\ 0 & \text{otherwise} \end{cases}$ // Equation (17)
        $\theta_k^i \leftarrow W \odot (\theta^{i-1} + m_k^i) + (1-W) \odot \theta^{i-1}$
    **end**

    **Server-side**
    **for** each client $k$ **do**
        $M_k^i \leftarrow \|\theta_k^i\|_c$ // Consensus
        $V_k^i \leftarrow \frac{\theta_k^i}{\|\theta_k^i\|_c}$ // Divergence
    **end**
    $M^i \leftarrow M^{i-1} + \frac{1}{N}\sum_{k=1}^{N}(M_k^i - M^{i-1})$ // Equation (6)
    **for** each vector slice $v$ in $V_k^i$ **do**
        $S_{pq} \leftarrow \cos(v_p, v_q)$
        $s_k \leftarrow \frac{1}{N-1}\sum_{j \neq k} S_{kj}$ // Average similarity
        $w_k \leftarrow \frac{\exp(s_k/\tau)}{\sum_{j=1}^{N}\exp(s_j/\tau)}$ // Softmax weights
    **end**
    $V^i \leftarrow V^{i-1} + \sum_{k=1}^{N} w_k(V_k^i - V^{i-1})$ // Equation (12)
    $\theta^i \leftarrow M^i \odot V^i$
**end**
**return** $\theta^T$

---

### 3.3. Importance-Aware Updating

**Motivation.** Conventional approaches require uploading all local LoRA parameters after each training round. However, during the SFT training process, the significance of parameter updates varies across different downstream tasks, with some parameters showing minimal significance. These unimportant parameters not only increase computational overhead during global aggregation but also can lead to catastrophic forgetting, degrading generalization performance (Dou et al., 2024; Yang et al., 2024; Han et al., 2024; Zhu et al., 2024; Liang et al., 2025; Huang et al., 2025). By selectively updating parameters for global model according to the importance, we can enhance model generalization during iterative SFT.

Table 1: Comparison with the state-of-the-art methods, providing by framework (Ye et al., 2024) [Arxiv24] for federated large language model fine-tuning, on different evaluation benchmark described in Appendix C. The best and second results are highlighted with bold and underline, respectively. Each value is accompanied by the improvement points compared to the benchmark model.

| Methods | Generalization [Arxiv23] | | | Code [Arxiv21] | FIN. [Arxiv20] | Math [Arxiv21] | Average |
| | MT-1 | MT-2 | MT-Final | Score | Score | Score | Rank |
|---|---|---|---|---|---|---|---|
| Base | 2.92 | 2.05 | 2.48 | 0.02 | 0.30 | 0.04 | 8 |
| FedAvg (McMahan et al.) | 4.58↑56.8% | 3.03↑47.8% | 3.80↑53.2% | 0.05↑250% | 0.35↑16.7% | 0.05↑25% | 4.6 |
| FedProx (Li et al.) | 4.29↑46.9% | 3.30↑61.0% | 3.79↑52.8% | 0.08↑400% | 0.28 | 0.13↑225% | 4.6 |
| FedAvgM (Hsu et al.) | 4.52↑54.8% | 2.95↑43.9% | 3.74↑50.8% | 0.09↑450% | **0.40**↑33.3% | 0.10↑250% | 4.0 |
| Scaffold (Karimireddy et al.) | 4.58↑56.8% | 3.10↑51.2% | 3.84↑54.8% | **0.10**↑500% | 0.30 | 0.12↑300% | 3.4 |
| FedAdam (Reddi et al.) | 4.45↑52.4% | 2.91↑42.0% | 3.67↑48.0% | 0.09↑450% | 0.38↑26.7% | 0.12↑300% | 4.2 |
| FedYogi (Reddi et al.) | 4.46↑52.7% | 2.90↑41.5% | 3.67↑48.0% | 0.06↑300% | 0.38↑26.7% | **0.16**↑400% | 4.6 |
| Ours (FedICU) | **4.83**↑65.4% | **3.43**↑67.3% | **4.13**↑66.5% | **0.10**↑500% | 0.35↑16.7% | 0.14↑350% | **1.8** |

**Importance-Aware Parameter Selection** For client $k$ at $i$-th iteration, where local parameters are initialized from global parameters $\theta^{i-1}$ and updated to $\theta_k^i$ after training. We introduce a momentum-enhanced dual-metric importance assessment mechanism that captures both the model generalization and specialization capabilities.

Inspired by parameter pruning methods (Han et al., 2015) and lottery ticket hypothesis (Frankle & Carbin, 2018), we propose that the magnitude of weights in the global model indicates the importance of individual parameters. We quantify the generalization importance $I$ through normalized and scaled weight magnitudes of the training parameters from the global model from iteration $i - 1$, as shown below:

$$I = \text{Sigmoid}(\frac{|\theta^{i-1}| - \mu(|\theta^{i-1}|)}{\sigma(|\theta^{i-1}|) + \epsilon}), \quad (14)$$

where $\sigma(\cdot)$ denotes the standard deviation, and $\mu(\cdot)$ computes the mean. We use the term $\epsilon = 10^{-6}$ to ensure formula numerical stability.

To mitigate parameter oscillations between consecutive update rounds, we employ momentum-based tracking of local model updates. The momentum term $m_k^i$ for client $k$ at iteration $i$ is computed as:

$$m_k^i = \beta m_k^{i-1} + (1 - \beta)(\theta_k^i - \theta^{i-1}), \quad (15)$$

where $\beta$ is the momentum coefficient, and $m_k^i$ represents the momentum term at iteration $i$.

We then define the specialization importance $G$ to quantifying the degree of local adaptation by measuring the relative magnitude of these momentum-smoothed updates, normalized using statistics from the previous iteration:

$$G = \text{Sigmoid}(\frac{|m_k^i| - \mu(|m_k^{i-1}|)}{\sigma(|m_k^{i-1}|) + \epsilon}). \quad (16)$$

During the client update process, parameters are selected for global model updates when their specialization importance exceeds their generalization importance. The parameter selection is determined by Equation (17). Note that in the

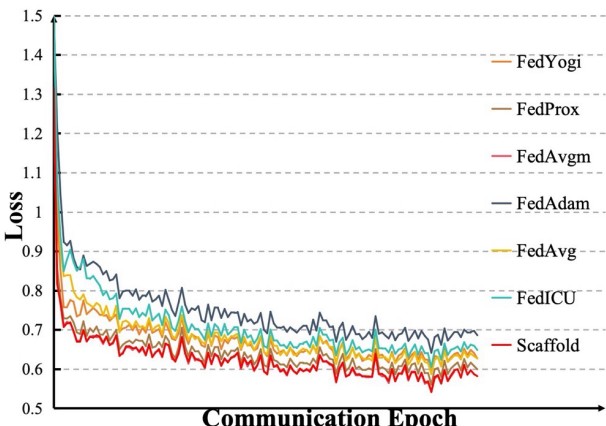

Figure 4: Different methods' training loss in Ye et al. (2024) benchmark of different epochs.

first iteration, all local model parameters are uploaded to initialize the aggregation process.

$$W[v] = \begin{cases} 1 & \text{if } G[v] > I[v], \\ 0 & \text{otherwise.} \end{cases} \quad (17)$$

The sparse parameter update is implemented using Equation (18). This selective update strategy prevents overfitting to the downstream instruction distribution of any specific client.

$$\theta_k^{i\prime} = W \odot (\theta^{i-1} + m_t) + (1 - W) \odot \theta^{i-1}, \quad (18)$$

where $\odot$ represents element-wise multiplication, and $\theta_k^{i\prime}$ is the model to be uploaded.

**Convergence Analysis.** We provide theoretical guarantees for the convergence of our proposed algorithm. The details can be found in the Appendix I.

## 4. Experiment

### 4.1. Experimental Setup

**Datasets**. Following Ye et al. (2024), we train our model on the four datasets Taori et al. (2023), Xiang Yue (2023),

Chaudhary (2023) and Yang et al. (2023b). We provide a details introduction in Appendix A.

**Framework Setup**. We conduct experiments using Nous-Research Llama-2-7b-hf model over 200 rounds. The model performance is evaluated using the benchmark metrics provided in Ye et al. (2024).

**Counterparts**. We compare our framework FedICU with several state-of-the-art methods (Ye et al., 2023) in the benchmark Ye et al. (2024): (1) Base Model without SFT. (2) FedAvg (McMahan et al., 2017b). (3) FedProx. (Li et al., 2018). (4) Scaffold (Karimireddy et al., 2019).(5) FedAvgM (Hsu et al., 2019). (6) FedAdam (Reddi et al., 2020). (7) FedYogi (Reddi et al., 2020). We provide a details introduction of different methods in Appendix A.

**Implement Details.** We present our experimental setup from three aspects:

- **Dataset Split**: Each client is assigned one of the four domain-specific datasets described in Appendix A, randomly sampling 5000 labeled examples for fine-tuning.

- **Training Setting**: We follow the benchmark Ye et al. (2024) default hyperparameters' configuration to set up our experiment environment. We provide a details introduction in Appendix D. All experiments are repeated three times to ensure statistical significance.

- **Evaluation Metric**: (1) Generalization: the first turn score from MT-Bench (Zheng et al., 2023) (2) Contextual Understanding: the final score from MT-Bench. (3) Code: Human Eval (Chen et al., 2021) (4) Financial: MMLU benchmark (Hendrycks et al., 2020) (5) Math: The GSM8k benchmark (Cobbe et al., 2021). We provide a details introduction in Appendix C.

**Performance Comparison.** Table 1 shows the performance comparison, while Figure 4 illustrates the training loss. Our results demonstrate FedICU superior performance in federated LLM training. Traditional methods such as FedAvg and FedProx struggle to effectively aggregate client consensus and align divergences, leading to the degradation of model performance. In contrast, FedICU successfully maintains the model generalization capabilities under these conditions while achieving superior performance across diverse downstream test sets. Specifically, our framework demonstrates a significant improvement in generalization and consistently maintains an advantage across various specialized domains. We also conduct more experiment on the different model architecture in Appendix E.

### 4.2. Diagnostic Experiments

**Ablation of Key Components.** We evaluate the contribution of each component using MT-Bench across four diverse datasets with optimized hyperparameters. Table 2 presents

Table 2: Ablation study of two key components mentioned in the passage, Consensus-Divergence Splitting (*Splitting* column in the table) and Important-Aware Updating (*Updating* column in the table). The best results are highlighted in bold.

| Updating | Splitting | Generalization | | |
|---|---|---|---|---|
| | | MT-1 | MT-2 | Final |
| ✗ | ✗ | 4.58 | 3.03 | 3.80 |
| ✓ | ✗ | 4.55 | 3.25 | 3.90 |
| ✗ | ✓ | 4.70 | 3.41 | 4.06 |
| ✓ | ✓ | **4.83** | **3.43** | **4.13** |

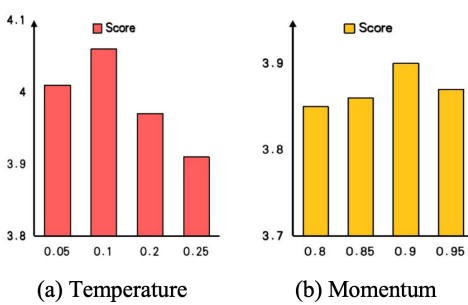

(a) Temperature     (b) Momentum

Figure 5: **Hyper-parameter evaluation** on MT-Bench, focusing on (a) the temperature $\tau$ in Equation (11) of Consensus-Divergence Splitting and (b) the momentum coefficient $\beta$ in Equation (15) of Importance-Aware Updating. Higher scores indicate better performance.

the results, demonstrating that both components contribute to performance improvements, with their combination yielding optimal results.

**Hyperparameters Study.** We analyze the sensitivity of the key hyperparameters of our method using MT-Bench, as shown in Figure 5. Results indicate that model performance decreases with increasing temperature above $0.1$, though remaining relatively stable within a certain range. The momentum factor shows minimal impact at higher values, demonstrating the robustness of our approach. Based on these findings, we adopt a temperature of $\tau = 0.1$ and a momentum factor of $\beta = 0.9$ in our experiments, as they perform best in the hyperparameters study.

**Parameter Distribution in FL LLM.** To demonstrate the rationale behind the Important-Aware Updating component, we conduct a study on the importance of the covered parameters shown in Table 3. The results show that Important-Aware Updating component physical significance. In the table, the parameter High-I remains stable while decreasing the parameter High-G proportions indicate better identification of core domain parameters. Declining overlap suggests that the Important-Aware Updating component has clearer functional partitioning, effectively separating general and domain-specific knowledge.

**The Efficient of the Binary Mask.** We conduct an ablation study to investigate the impact of momentum-based param-

Table 3: Parameter distribution in FL LLM. Overlap indicates important parameters globally and locally, High-I represents parameters emphasizing general ability, and High-G represents parameters emphasizing domain capability.

| Class/Round | 5 | 15 | 25 | 35 | 45 | 55 | 65 |
|---|---|---|---|---|---|---|---|
| Overlap | 0.24 | 0.16 | 0.13 | 0.12 | 0.11 | 0.10 | 0.08 |
| High-I | 0.48 | 0.46 | 0.46 | 0.45 | 0.45 | 0.44 | 0.44 |
| High-G | 0.50 | 0.40 | 0.38 | 0.38 | 0.37 | 0.36 | 0.35 |

Table 4: Experiment results of ablation study about Importance-Aware Update. MT-1 shows the model general capability, while MT-2 shows the model level of contextual understanding. MT-Final is the metric combining both of the two indicators.

| Momentum | Smooth | MT-1 | MT-2 | MT-Final |
|---|---|---|---|---|
| ✗ | ✓ | 4.59 | 3.20 | 3.90 |
| ✓ | ✗ | 4.60 | 3.33 | 3.97 |
| ✓ | ✓ | 4.65 | 3.37 | 4.01 |
| ✓ (Ours) | ✗ (Ours) | **4.83** | **3.43** | **4.13** |

eter selection and continuous-valued parameter weighting on model generalization capability, as shown in the table below Table 4. In the Momentum column, we indicate whether we include the momentum component to smooth the mask construction process. In the Smooth column, we use a smoothing mask approach, when $G[v] > I[v]$ making $W[v] = \min\left(1, \frac{G[v]}{I[v]+G[v]}\right)$, to filter uploaded parameters and their weights. The results validate that Important-Aware Updating component is a simple and effective method.

### 4.3. Communication Cost

Regarding FedICU, since LoRA splitting and merging can be performed locally on either the client or server, it doesn't introduce additional communication overhead compared to standard federated learning (Ye et al., 2025b;a). For the Importance-Aware Updating component, assuming there are $K$ clients, each with $N$ parameters, and in the mask built by the parameter importance selection upload component, the proportion that needs to be uploaded is $\alpha$. So the communication cost of standard federated learning would be $C_{std} = O(K * N)$, while the communication cost of FedICU is $C_{imp} = O\left(\sum_{k=1}^{K} N \times \alpha_k\right)$, where $\alpha_k < 1$, which demonstrates the communication savings of FedICU.

## 5. Discussion and Limitation

(1) While our method leverages LoRA, which may not achieve the same performance levels as full parameter fine-tuning. Despite these trade-offs, LoRA significantly reduces computational requirements and enables rapid adaptation to diverse downstream tasks. Future work could focus on bridging this performance gap while maintaining LoRA computational advantages.

(2) Although our method is primarily developed for fed-

erated large language model training, the underlying principles of our approach have broader implications. Future research could investigate the application of these concepts to other domains within federated learning, potentially leading to new insights and methodological advances.

## 6. Conclusion

In this paper, we introduce FedICU, a framework that employs consensus-divergence splitting for effective LLM fine-tuning in federated settings. Our method decouples LoRA updates into divergence and consensus components, enabling fine-grained control over model updates. At the server level, our approach aligns divergence using cosine similarity and aggregates consensus, significantly improving the global model generalization ability. At the client level, we implement an importance-aware updating method that selectively updates parameters, preventing overfitting and preserving the global model generalization capabilities. Extensive experiments demonstrate the effectiveness and robustness of our approach. We believe this work provides valuable insights for future research in federated large language model fine-tuning, particularly for heterogeneous downstream tasks.

## Acknowledge

This research was financially supported by the Open Research Fund from Guangdong Laboratory of Artificial Intelligence and Digital Economy (SZ), under Grant No. GML-KF-24-10, the National Natural Science Foundation of China under Grant (62361166629, 62176188, 62225113, 623B2080), the National Key Research and Development Program of China (2024YFC3308400), the Wuhan University Undergraduate Innovation Research Fund Project, and the Hubei Postdoctoral Talent Introduction Program (2024HBBHJD070). The supercomputing system at the Supercomputing Center of Wuhan University supported the numerical calculations in this paper.

## Impact Statement

This paper presents work whose goal is to advance the field of Machine Learning. There are many potential societal consequences of our work, none which we feel must be specifically highlighted here.

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

## A. Datasets

- Taori et al. (2023): The dataset used for fine-tuning the Alpaca model. Alpaca is a dataset of 52,000 instructions and demonstrations generated by OpenAI text-davinci-003 engine.

- Xiang Yue (2023): The dataset concerning the math field. Math instruct is compiled from 13 math rationale datasets, six of which are newly curated by this work. It uniquely focuses on the hybrid use of chain-of-thought (CoT) and program-of-thought (PoT) rationales, and ensures extensive coverage of diverse mathematical fields.

- CodeAlpaca-20k: The dataset concerning the code field. The 20K instruction-following data generated by the techniques Self-Instruct (Wang et al., 2022), with some modifications by author of the datasets.

- FinGPT: The specialized financial datasets used in FinGPT (Yang et al., 2023b).

## B. Counterparts

(1) **Base Model without SFT**.

(2) **FedAvg** (McMahan et al., 2017b). The standard federated averaging algorithm, where updates from all clients are averaged at the server.

(3) **FedProx** (Li et al., 2018). An extension of FedAvg that introduces a proximal term to tackle heterogeneity across clients.

(4) **Scaffold** (Karimireddy et al., 2019). A control variate-based method designed to reduce the impact of client drift in federated learning with non-IID data.

(5) **FedAvgM** (Hsu et al., 2019). A momentum-based variant of FedAvg, which integrates server-side momentum into the federated learning process.

(6) **FedAdam** (Reddi et al., 2020). A federated version of the Adam optimizer. It adapts the learning rates at the server side using first and second-order moments of gradients, aiming to provide better performance in challenging federated settings.

(7) **FedYogi** (Reddi et al., 2020). An adaptive federated optimization method that improves robustness and convergence in federated learning, which effectively handles the challenges of non-IID data and system heterogeneity in federated settings.

## C. Evaluation Metric

(1) **Generalization**: We use the first turn score from MT-Bench (Zheng et al., 2023) as the primary evaluation metric to assess the general performance of different models (Ye

Table 5: The performance of FedICU and other methods applied in the Mistral-7B. MT-1 shows the model general capability, while MT-2 shows the model level of contextual understanding. MT-Final serves as a comprehensive metric that combines both of the aforementioned indicators.

| Method | MT-1 | MT-2 | MT-Final |
|--------|------|------|----------|
| Base | 4.10 | 3.29 | 3.69 |
| FedAvg | 5.48 | 3.80 | 4.64 |
| Scaffold | 5.46 | 3.92 | 4.69 |
| FedYogi | 5.54 | 3.85 | 4.70 |
| FedProx | 5.50 | 3.83 | 4.67 |
| FedAdam | 5.56 | 3.91 | 4.73 |
| FedAvgM | 5.55 | 3.73 | 4.64 |
| Ours | **5.58** | **4.11** | **4.84** |

et al., 2024). This score is the most critical in the overall evaluation.

(2) **Contextual Understanding**: We use the final score from MT-Bench to evaluate the model ability to understand context. MT-Bench comprises two turns of conversation, making it suitable for contextual testing.

(3) **Code**: We use Human Eval (Chen et al., 2021) to evaluate the model coding capabilities.

(4) **Financial**: We utilize the MMLU benchmark (Hendrycks et al., 2020) to evaluate the model financial knowledge, specifically selecting the finance-related domains for this assessment.

(5) **Math**: The GSM8k benchmark (Cobbe et al., 2021) is used to test the model mathematical abilities.

## D. Training Settings

For all experiments, we use the following hyperparameters settings:

- Learning rate: $5e-5$

- Training rounds: 200

- FedProx proximal term $\mu$: 0.01

- FedYogi and FedAdam:

  - Server learning rate: $1e-3$
  - $\tau$: $1e-3$

- FedAdam momentum coefficient $\beta$: 0.9

## E. More Experiments on Different Model Architecture

We conduct supplementary experiments with Mistral-7B to verify that our method is effective across a broader range of

Table 6: The $\mu$ and $\theta$ of consensus parameters' delta update. $\mu$ represents the mean of the consensus vector updates, and $\theta$ represents the fluctuations during the consensus update process.

| Client | $\mu$ | $\theta$ |
|--------|-------|----------|
| 1 | 1.452974 | $1.53*10^{-4}$ |
| 2 | 1.452972 | $1.54*10^{-4}$ |
| 3 | 1.452974 | $1.56*10^{-4}$ |
| 4 | 1.452973 | $1.54*10^{-4}$ |

model architectures. The results of these supplementary experiments are in the Table 5. The results show that FedICU also demonstrates excellent performance across models with different architectures.

## F. The experiment on consensus noise updating action.

In Figure 3, we use cosine similarity to support the assumption that "consensus noise" is the same. To further support it, we conduct an experiment to explore the mean and variation of the consensus vector updates for each client under the condition of independent updates across clients, as shown in the table Table 6. From the table, the similarity of $\mu$ and $\theta$ across different clients indicates that the consensus noise is consistent.

## G. Splitting Convergence Analysis

Let $F(\theta)$ denote the global objective function that measures the model performance across all clients, where $\theta \in \mathbb{R}^d$ represents the global model parameters. We prove that our *Consensus-Divergence Splitting* method converges under standard assumptions.

**Assumption G.1** (Smoothness). $F(\theta)$ is $L$-smooth: for all $\theta, \theta' \in \mathbb{R}^d$:

$$\|\nabla F(\theta) - \nabla F(\theta')\| \leq L\|\theta - \theta'\|. \quad (19)$$

**Assumption G.2** (Bounded Gradient). There exists $G > 0$ for all $\theta$:

$$\|\nabla F(\theta)\| \leq G. \quad (20)$$

**Assumption G.3** (Unbiased & Bounded-Variance Gradient Estimates). During round $i$, each client $k$ produces update $\Delta_k^i = \theta_k^i - \theta^i$. The local gradient estimator is:

$$\widetilde{g}_k^i = -\frac{1}{\eta^i}\Delta_k^i. \quad (21)$$

where $\eta^i$ is the server step size. We assume:

**Unbiasedness:**

$$\mathbb{E}\left[\sum_{k=1}^{N} w_k^i \widetilde{g}_k^i \mid \theta^i\right] = \nabla F(\theta^i). \quad (22)$$

where $w_k^i \geq 0, \sum_k w_k^i = 1$ are similarity-based weights.

**Bounded variance:** For $\sigma^2 \geq 0$:

$$\mathbb{E}[\|\widetilde{g}_k^i - \nabla F(\theta^i)\|^2] \leq \sigma^2. \tag{23}$$

**Assumption G.4** (Strong Convexity (Optional)). $F(\theta)$ is $\mu$-strongly convex ($\mu > 0$): for all $\theta, \theta' \in \mathbb{R}^d$:

$$F(\theta') \geq F(\theta) + \langle \nabla F(\theta), \theta' - \theta \rangle + \frac{\mu}{2}\|\theta' - \theta\|^2. \tag{24}$$

Note that Assumption G.4 is optional. If $F$ is not strongly convex (e.g., just convex or nonconvex), we prove convergence to a stationary point. If $F$ is strongly convex, we show convergence to the unique global optimum.

At iteration $i$, FedICU aggregates local models via:

$$\theta^{i+1} = \theta^i + \sum_{k=1}^{N} w_k^i(\theta_k^i - \theta^i) = \theta^i - \eta^i \sum_{k=1}^{N} w_k^i \widetilde{g}_k^i. \tag{25}$$

We employ cosine decay learning schedule:

$$\eta^i = \eta_{\max} \cdot \frac{1 + \cos(\frac{\pi i}{T_{\max}})}{2}. \tag{26}$$

**Theorem G.5** (Unified Convergence of FedICU). *Under Assumptions G.1–G.3, let $\{\theta^i\}$ be produced by (25). Then:*

*For smooth possibly nonconvex $F$:*

$$\min_{0 \leq i < T} \mathbb{E}[\|\nabla F(\theta^i)\|^2] \leq O(\frac{1}{\sqrt{T}}). \tag{27}$$

*For $\mu$-strongly convex $F$, FedICU converges to the unique global optimum $\theta^*$ with rate depending on $\mu$, $L$, $\sigma^2$.*

*Proof.* By $L$-smoothness:

$$F(\theta^{i+1}) \leq F(\theta^i) + \langle \nabla F(\theta^i), \theta^{i+1} - \theta^i \rangle + \frac{L}{2}\|\theta^{i+1} - \theta^i\|^2. \tag{28}$$

Using update rule and bounded-variance assumption:

$$\min_{0 \leq i < T} \mathbb{E}[\|\nabla F(\theta^i)\|^2] \leq O(\frac{1}{\sum_{i=0}^{T-1} \eta^i}) + O(\sum_{i=0}^{T-1}(\eta^i)^2). \tag{29}$$

When $\sum_{i=0}^{T-1} \eta^i = \Omega(\sqrt{T})$ and $\sum_{i=0}^{T-1}(\eta^i)^2 = O(1)$, we get $O(1/\sqrt{T})$ rate. Under strong convexity, additional terms yield $\|\theta^i - \theta^*\| \to 0$. □

For finite-$T$ training with cosine decay schedule, our analysis shows that the model converges as the learning rate naturally decays to near-zero at the end of training.

## H. Optimality of Averaging Strategy

Here we prove that direct averaging of consensus vectors yields the optimal linear unbiased estimator. We formalize the problem as follows: Given the consensus vector $M_k^i$ of client $k$ at iteration $i$:

$$M_k^i = M^* + \xi_k, \tag{30}$$

where $M^*$ represents the true global consensus and $\xi_k \sim \mathcal{N}(0, \sigma^2 I)$ denotes the noise term. For any linear unbiased estimator $\hat{M} = \sum_{k=1}^{N} \alpha_k M_k^i$ with the constraint $\sum_{k=1}^{N} \alpha_k = 1$, its variance is:

$$\text{Var}(\hat{M}) = \sum_{k=1}^{N} \alpha_k^2 \sigma^2. \tag{31}$$

To find the optimal weights $\alpha_k$ that minimize the variance, we use the Lagrange multiplier method. The Lagrangian is:

$$L(\alpha_1, \ldots, \alpha_N, \lambda) = \sum_{k=1}^{N} \alpha_k^2 \sigma^2 + \lambda(\sum_{k=1}^{N} \alpha_k - 1). \tag{32}$$

Taking partial derivatives:

$$\frac{\partial L}{\partial \alpha_k} = 2\alpha_k \sigma^2 + \lambda = 0 \quad \frac{\partial L}{\partial \lambda} = \sum_{k=1}^{N} \alpha_k - 1 = 0. \tag{33}$$

From the first equation:

$$\alpha_k = -\frac{\lambda}{2\sigma^2}. \tag{34}$$

This shows all $\alpha_k$ are equal. Combined with the constraint:

$$\sum_{k=1}^{N} \alpha_k = N\alpha_k = 1. \tag{35}$$

Therefore:

$$\alpha_k = \frac{1}{N}, \quad k = 1, 2, \ldots, N. \tag{36}$$

The second derivative:

$$\frac{\partial^2 L}{\partial \alpha_k^2} = 2\sigma^2 > 0. \tag{37}$$

confirms this is indeed a minimum. This proof demonstrates that equal weights $\alpha_k = \frac{1}{N}$ minimize the variance of the linear unbiased estimator, justifying our averaging strategy:

$$M^i = M^{i-1} + \frac{\sum_{k=0}^{N}(M_k^i - M^{i-1})}{N}. \tag{38}$$

# I. Updating Convergence Analysis

We analyze the convergence of our *Importance-Aware Updating* mechanism. Following the same assumptions and notations in Appendix G, let $\theta \in \mathbb{R}^d$ denote the global model parameters and $F(\theta)$ be the global objective function:

$$F(\theta) = \frac{1}{N} \sum_{k=1}^{N} F_k(\theta), \qquad (39)$$

where $F_k$ is the local loss at client $k$.

For parameter selection and updates, we define importance metrics for each parameter $v$:

$$I[v] = \text{Sigmoid}\Big(\frac{|\theta^{i-1}[v]| - \mu(|\theta^{i-1}|)}{\sigma(|\theta^{i-1}|) + \epsilon}\Big). \qquad (40)$$

$$G[v] = \text{Sigmoid}\Big(\frac{|m_k^i[v]| - \mu(|m_k^{i-1}|)}{\sigma(|m_k^{i-1}|) + \epsilon}\Big). \qquad (41)$$

The binary mask $W \in \{0, 1\}^d$ determines which parameters to update, as the formula Equation (42):

$$W[v] = \begin{cases} 1, & \text{if } G[v] > I[v], \\ 0, & \text{otherwise.} \end{cases} \qquad (42)$$

Each client uploads a masked model:

$$\theta_k'^i = W \odot (\theta^{i-1} + m_k^i) + (1 - W) \odot \theta^{i-1}. \qquad (43)$$

We can rewrite this update as:

$$\theta_k'^i = \theta^{i-1} + W \odot m_k^i = \theta^{i-1} + m_k^i - \Delta_k^i, \qquad (44)$$

where $\Delta_k^i = (1 - W) \odot m_k^i$ represents masked-out updates.

At iteration $i$, the server aggregates:

$$\theta^i = \frac{1}{N} \sum_{k=1}^{N} \theta_k'^i = \theta^{i-1} + \frac{1}{N} \sum_{k=1}^{N} (m_k^i - \Delta_k^i). \qquad (45)$$

We employ cosine decay learning schedule as in the previous section:

$$\eta^i = \eta_{\max} \cdot \frac{1 + \cos(\frac{\pi i}{T_{\max}})}{2}. \qquad (46)$$

**Theorem I.1** (Convergence). *Under Assumptions G.1–G.3, let $\{\theta^i\}$ be generated by the masked update rule. If the masking error is bounded:*

$$\mathbb{E}[\|\frac{1}{N} \sum_{k=1}^{N} \Delta_k^i\|^2] \leq \delta^2. \qquad (47)$$

*Then for smooth $F$:*

$$\min_{0 \leq i < T} \mathbb{E}[\|\nabla F(\theta^i)\|^2] \leq O(\frac{1}{\sqrt{T}}). \qquad (48)$$

*For $\mu$-strongly convex $F$, convergence to the unique global optimum $\theta^*$ is achieved with rate depending on $\mu$, $L$, $\sigma^2$, and $\delta$.*

*Proof.* By $L$-smoothness:

$$F(\theta^i) \leq F(\theta^{i-1}) + \langle \nabla F(\theta^{i-1}), \theta^i - \theta^{i-1}\rangle + \frac{L}{2}\|\theta^i - \theta^{i-1}\|^2. \qquad (49)$$

Substituting the masked update:

$$\theta^i - \theta^{i-1} = \frac{1}{N} \sum_{k=1}^{N} (m_k^i - \Delta_k^i). \qquad (50)$$

The first term averages to the true gradient (by unbiasedness), while the masking error $\Delta_k^i$ is bounded by $\delta^2$. Following the analysis in Appendix G, this additional error term contributes only a constant factor to the convergence rate:

$$\min_{0 \leq i < T} \mathbb{E}[\|\nabla F(\theta^i)\|^2] \leq O(\frac{1}{\sum_{i=0}^{T-1} \eta^i}) + O(\sum_{i=0}^{T-1} (\eta^i)^2) + O(\delta^2). \qquad (51)$$

With cosine decay schedule, when $\sum_{i=0}^{T-1} \eta^i = \Omega(\sqrt{T})$ and $\sum_{i=0}^{T-1} (\eta^i)^2 = O(1)$, we achieve the $O(1/\sqrt{T})$ rate. For the strongly convex case, the additional error term similarly affects only the constants in the convergence rate to $\theta^*$. $\square$

These results show that our importance-aware parameter updating maintains convergence guarantees while allowing adaptive sparsification of updates.

