# OpenReview forum: "Splitting with Importance-aware Updating for Heterogeneous Federated Learning with Large Language Models"
_ICML.cc/2025/Conference — ICML 2025 poster_

### Official Review · Reviewer_CCAN · 2025-03-09

**Overall Recommendation:** 4

**Summary:**

The manuscript addresses federated LLM fine-tuning, highlighting how existing methods often lead to catastrophic forgetting, diminishing the global model's generality, and failing to properly balance model updates across clients with different downstream task datasets. The authors propose the FedICU method, which cleverly enhances the global model's generalization capabilities through importance-aware parameter uploading and consensus-divergence splitting. Experimental results demonstrate the effectiveness of the authors' proposed framework.

**Claims And Evidence:**

The claims made in this submission are well-supported by compelling evidence.
1. The authors provide detailed theoretical foundations for both key components in the framework. What's more, the authors provide comprehensive and accurate convergence analysis to elucidate the correctness of the theory.
2. The authors conducted training on heterogeneous datasets and provided complete and detailed test results. Experiments show that the proposed framework outperforms state-of-the-art methods on multiple test benchmarks, demonstrating significant performance improvements.

**Essential References Not Discussed:**

No

**Experimental Designs Or Analyses:**

This paper’s experimental design is robust, comparing FedICU against the state-of-the-art methods across different evaluation criteria. The ablation studies effectively also isolate the contribution of each component.

**Methods And Evaluation Criteria:**

The lack of public datasets has made federated LLM fine-tuning an innovative direction. This paper effectively addresses the issue of decreased global model generalization capability during federated large model fine-tuning due to dataset heterogeneity, providing a novel solution for the field.

**Other Comments Or Suggestions:**

No

**Other Strengths And Weaknesses:**

Strengths:
1. Studies on combining federated learning with large language models are crucial given the growing scarcity of public training data and increasing demand for privacy-preserving distributed training. This work addresses a critical challenge in this emerging field by tackling non-IID instruction scenarios.
2. The proposed consensus-divergence splitting mechanism and importance-aware updating strategy work synergistically to balance global model capabilities with domain adaptation. The theoretical foundation is solid with comprehensive convergence analysis, and the implementation is elegant with clear architectural design.
3. The framework's approach to decomposing and balancing model updates has broader implications beyond LLMs. This framework offers new insights for federated learning in other domains where maintaining general capabilities while enabling specialization is crucial.

Weakness:
1. The paper lacks explanation regarding the communication efficiency of the proposed method.

**Questions For Authors:**

1. Is the proposed framework orthogonal to other LoRA training methods?
2. The Importance-aware Updating mechanism focuses on parameter selection based on importance. Have the authors explored the communication efficiency benefits this might provide?

**Relation To Broader Scientific Literature:**

The authors focus on addressing the growing challenge of enhancing global model generalization when fine-tuning LLMs in distributed heterogeneous environments. This work is situated within the broader context of diminishing high-quality public training datasets for LLMs and the increasing adoption of federated learning approaches. By building on federated learning foundations, the paper specifically tackles the problems arising from heterogeneous data distributions across clients when using LoRA for LLM fine-tuning

**Theoretical Claims:**

The paper provides theoretical analysis for both components of their framework. The proofs for convergence analysis (Sections E, F, and G in the appendix) establish the mathematical foundations for their approach.

---

> ### Author Rebuttal · Authors · 2025-03-31
>
> Dear Reviewer CCAN:
>
> Thank you very much for your recognition of our work. Below I will provide detailed responses to your questions. Thank you for taking your valuable time to offer suggestions for our work.
>
> **Q1:Is the proposed framework orthogonal to other LoRA training methods?** (Questions For Authors)
>
> **A1:** Yes, our method can be divided into two components, applied to both client and server parts.
> On the client side, we select important parameters based on generalization importance and special importance, and perform sparse updates on these important parameters. This is compatible with existing LoRA training methods since this process doesn't affect the training logic but simply adds a masking module.
> On the server side, we split and aggregate LoRA parameters, which is equivalent to first splitting LoRA into two parameter matrices on the server, applying the same training logic to both matrices separately, and then aggregating them back into a single LoRA parameter matrix at the end. Therefore, it doesn't interfere with the LoRA training process either, as it can be considered as only modifying the input/output pipeline stage of the server's LoRA training.
> In conclusion, our method is orthogonal to other methods for training LoRA.
>
> **Q2: Have the authors explored the communication efficiency benefits this might provide?** (Other Strengths And Weaknesses & Questions For Authors)
>
> **A2:** Thank you for your suggestions. Regarding FedICU, since LoRA splitting and merging can be performed locally on either the client or server, it doesn't introduce additional communication overhead compared to standard federated learning. For the Importance-Aware Updating component, we'll analyze it mathematically.
> Assuming there are $K$ clients, each with $N$ parameters, and in the mask built by the parameter importance selection upload component, the proportion that needs to be uploaded is $\alpha$. So the communication cost of standard federated learning would be $C_{std} = O(K * N)$ , while the communication cost of our method is $C_{imp} = O\left(\sum_{k=1}^{K} N \times \alpha_k\right), \text{ where } \alpha_k < 1$, which demonstrates the communication savings of our method.

---

### Official Review · Reviewer_UVcT · 2025-03-12

**Overall Recommendation:** 3

**Summary:**

This paper proposes FedICU to address client heterogeneity in FL. FedICU consists of two key components:

1. **Consensus-Divergence Splitting**: This method decomposes client updates into magnitude and direction, treating magnitude as consensus and direction as divergence. The two components are then aggregated separately.
2. **Importance-Aware Parameter Selection**: This technique selects the most important parameters for sparse updating, enhancing efficiency.

Numerical results demonstrate the superior performance of the proposed method compared to baseline approaches.


## update after rebuttal.

I remain unconvinced by the following points:
(1) The rationale for using direction (cosine similarity) to measure differences in magnitude is unclear.
(2) Whether the observation that "client updates tend to have similar magnitudes but vary in direction" sufficiently demonstrates that magnitudes represent consensus while directions indicate divergence remains unaddressed.
(3) Whether the proposed method preserves knowledge in scenarios *where base models perform well* is still not adequately explained.

However, I acknowledge and respect the efforts made by the authors, and I am increasing my rating to 3.

**Claims And Evidence:**

The claim regarding the relationship between magnitude-direction and consensus-divergence does not appear to be particularly strong. The authors have demonstrated that client updates tend to have similar magnitudes but vary in direction. However, it is unclear whether this observation is sufficient to substantiate their claim.

**Essential References Not Discussed:**

NA

**Experimental Designs Or Analyses:**

Currently, only the final performance on domain knowledge is reported, which may not be sufficient to fully support the key motivation of this paper. To strengthen the claims, please consider including the following experiments:

1. **Preservation of General Knowledge**: Demonstrate that FedICU does not disrupt the general knowledge of LLMs, whereas other baselines do. This can be achieved by evaluating performance on benchmarks where the backbone LLMs typically perform well.

2. **Effectiveness of Sparse Gradients**: Show that sparse gradients effectively prevent catastrophic forgetting while maintaining generalization performance.

3. **Convergence Improvement**: Provide evidence that Consensus-Divergence Splitting accelerates convergence compared to the baselines.

4. **Handling Client Drift**: Illustrate how FedICU excels in managing varying levels of client drift.

5. **Case Studies on Client Drift**: Include case studies where client drift occurs, highlighting instances where baseline methods fail while FedICU performs well.

**Methods And Evaluation Criteria:**

Some aspects of the method design are unclear:

1. **Notation Confusion**: The notation appears inconsistent. According to Eq. (1), we have \(\Delta \theta \in \mathbb{R}^{d \times k}\). However, before Eq. (5), this changes to \(\theta_i \in \mathbb{R}^{r \times d}\), and before Eq. (9), it becomes \(\Delta \theta_i^{k} \in \mathbb{R}^{d \times d}\). Could the authors clarify which part of the LoRA parameters is actually being updated?

2. **Assumption in Eq. (7)–(8)**: The design of Eq. (7)–(8) relies on the assumption that "the noise of the consensus is the same." The authors state that this is supported by the results in Figure 3. However, they do not specify which metric is used for Figure 3. If cosine similarity is used, Figure 3 may not sufficiently support this assumption. Could the authors provide further clarification?

**Other Comments Or Suggestions:**

Since LLMs are generative models, a key concern is whether FedICU could unintentionally expose user data when local LoRA updates are uploaded to the server. Could the authors clarify how FedICU prevents this risk?

**Other Strengths And Weaknesses:**

The paper is easy to follow.

**Questions For Authors:**

NA

**Relation To Broader Scientific Literature:**

NA

**Theoretical Claims:**

The convergence rates are given.

---

> ### Author Rebuttal · Authors · 2025-03-31
>
> Dear Reviewer UVcT:
>
> Thank you very much for your time and review suggestions on our paper. We hope the following responses can address your concerns.
>
> **Q1：The claim regarding the relationship between magnitude-direction and consensus-divergence does not appear to be particularly strong.** (Claims And Evidence)
>
> **A1:** During model training, parameters are optimized toward both global and local optima, and direct aggregation leads to decreased general ability. We decompose parameters into consensus and divergence, where the former represents similar update behaviors across clients, while the latter represents domain-specific update characteristics. We find that during training, magnitude (consensus) updates remain relatively consistent, while directional (divergence) updates differ significantly. This shows that clients maintain general capabilities through consensus parameters and domain-specific capabilities through divergence parameters across downstream tasks.
> To further support our claim, we conduct an experiment with results shown in the table below. We find that consensus are similar across clients, while divergence differ more significantly, supporting our argument. We will add this experiment to revised manuscript to further substantiate our claim.
>
> *Table: Similarity of consensus matrices.*
> |Clients|1|2|3|4|
> |-|-|-|-|-|
> |1|1|0.85|0.89|0.90|
> |2|0.85|1|0.85|0.91|
> |3|0.89|0.85|1|0.90|
> |4|0.90|0.91|0.90|1|
>
> *Table: Similarity of divergence matrices.*
> |Clients|1|2|3|4|
> |-|-|-|-|-|
> |1|1|0.23|0.49|0.04|
> |2|0.23|1|0.34|0.05|
> |3|0.49|0.34|1|0.11|
> |4|0.04|0.05|0.11|1|
>
> *Table: Similarity of unsplit matrices.*
> |Clients|1|2|3|4|
> |-|-|-|-|-|
> |1|1|0.32|0.54|0.05|
> |2|0.32|1|0.35|0.07|
> |3|0.54|0.35|1|0.14|
> |4|0.05|0.07|0.14|1|
>
> **Q2: Notation Confusion** (Methods And Evaluation Criteria)
>
> **A2:** We apologize for the inconsistent notation.
> - In Eq.1, $\Delta\theta \in \mathbb{R}^{d\times k}$.
> - For the others, the dimension of LoRA should be corrected to $d$ x $k$. In Eq.9 part, we introduce $A$ and $B$ as its decomposition matrices, with dimensions $d$ x $r$ and $r$ x $k$ respectively.
>
> **Q3: Assumption in Eq. 7, 8 is not fully supported by Figure 3.** (Methods And Evaluation Criteria)
>
> **A3:** In Figure 3, we use cosine similarity to support the assumption that "consensus noise" is the same. To further support it, we conduct an experiment, as shown in the table below.
>
> *Table: the $\mu$ and $\theta$ of consensus parameters' update. $\mu$ represents the mean of the consensus vector updates, and $\theta$ represents the fluctuations during the update process.*
> |Client|$\mu$|$\theta$|
> |-|-|-|
> |1|37.44|1.06|
> |2|37.44|1.06|
> |3|37.44|1.06|
> |4|37.44|1.06|
>
> From the table, the similarity of $\mu$ and $\theta$ across different clients indicates that the consensus noise is consistent.
>
> **Q4：Need experiments to strengthen the claims in different aspects.** (Experimental Designs Or Analyses)
>
> **A4:** Thank you for your advice. We discuss our experiment and conduct more to further support our claims below.
>
> - Preservation of General Knowledge: Table 1 in paper details our model's performance. We measure FedICU on MT-Bench[1] to test general capabilities. The optimal performance of ours demonstrates that it can preserve the model's general knowledge to a certain extent.
> - Effectiveness of Sparse Update: According to related research [2, 3], sparse updates can help prevent catastrophic forgetting. Additionally, Table 2 in our paper shows that the model improves when including Sparse Update, proving the effectiveness.
> - Convergence Improvement: We conduct an experiment measuring the rounds to reach average loss, with the table below. We can find that FedICU optimizes convergence.
> - Handling Client Drift & Case Studies: In Table 1 of our paper, we test FedICU with other methods across various domains. FedICU achieves optimal average domain knowledge, meaning it effectively handles domain knowledge without excessive bias.
>
> *Table: Convergence rounds of FedICU and other methods.*
> |Method|FedAvg|FedProx|FedAvgM|Scaffold|FedAdam|FedYogi|FedICU|
> |-|-|-|-|-|-|-|-|
> |Round|53|43|45|38|37|39|34|
>
> We will add this discussion and experiment in revised manuscript to support our claims.
>
> [1] Openfedllm: Training large language models on decentralized private data via federated learning. **KDD 2024.**
> [2] A simple and effective pruning approach for large language models. **arXiv 2023.**
> [3] LoRASculpt: Sculpting LoRA for Harmonizing General and Specialized Knowledge in Multimodal Large Language Models. **arXiv 2025.**
>
> **Q5: A key concern is whether FedICU could unintentionally expose user data to the server.** (Other Comments Or Suggestions)
>
> **A5:** FedICU follows FL protocols by only sharing parameter updates, not raw data. With Importance-Aware Update mechanism, FedICU can further reduce exposure risk by filtering parameters. Additionally FedICU follows standard FL logic and is compatible with other privacy-enhancing solutions.

---

### Official Review · Reviewer_UEdC · 2025-03-12

**Overall Recommendation:** 4

**Summary:**

The paper proposes FedICU, a novel federated learning framework for large language models (LLMs) in heterogeneous settings. It decomposes client updates into consensus and divergence components. In the global aggregation phase, it balances these components based on their contribution to the global model performance. At the client level, it uses an importance - aware parameter updating strategy. Experiments across various domains show that FedICU outperforms existing federated learning approaches in generalization performance and domain adaptation.

**Claims And Evidence:**

Yes.

**Essential References Not Discussed:**

There do not seem to be any essential references not discussed in the paper.

**Experimental Designs Or Analyses:**

The experimental designs are reasonable. The authors train the model on four different datasets (Taori et al., 2023; Xiang Yue, 2023; CodeAlpaca - 20k; FinGPT) to simulate heterogeneous data distribution. They compare FedICU with multiple state - of - the - art methods in the same experimental setting. The hyperparameters are set following a benchmark, and all experiments are repeated three times to ensure statistical significance. The ablation study of key components and hyperparameter study further validate the effectiveness and sensitivity of the proposed framework.

**Methods And Evaluation Criteria:**

The proposed methods and evaluation criteria make sense for the problem. The Consensus - Divergence Splitting method decomposes LoRA updates in a way that effectively captures general and client-specific knowledge, which is crucial for handling heterogeneous downstream instructions. The Importance - Aware Updating method reduces computational overhead and prevents catastrophic forgetting. For evaluation, using metrics like generalization (first turn’s score from MT - Bench), contextual understanding (final score from MT - Bench), and domain - specific metrics (e.g., Human Eval for code, MMLU for finance, GSM8k for math) comprehensively assesses the model's performance in different aspects relevant to LLMs in federated learning.

**Other Comments Or Suggestions:**

Please refer to "Other Strengths And Weaknesses" box.

**Other Strengths And Weaknesses:**

A major strength of the paper is its innovative framework, which effectively addresses the challenges in heterogeneous federated learning for LLMs. The combination of Consensus - Divergence Splitting and Importance - Aware Updating is novel and shows significant performance improvement. However, a weakness is that the method relies on LoRA, which may not achieve the same performance as full - parameter fine - tuning. Also, the experimental results are mainly based on specific datasets and models, and the generalization to other scenarios may need further investigation.

**Questions For Authors:**

1. In the Importance - Aware Updating mechanism, the threshold for parameter selection (comparing generalization and specialization importance) is based on a binary decision. Have you considered using a more flexible thresholding method, and how would it affect the performance? A more flexible method might better balance the trade-off between preserving general knowledge and adapting to specific domains. If it leads to a significant improvement in performance, it could strengthen the proposed approach.
2. The experiments are conducted on specific datasets and a single model (Llama-2-7b-hf). How do you expect the performance of FedICU to scale when applied to larger models or different model architectures? If FedICU can show consistent performance improvement across different model scales and architectures, it would greatly expand its applicability.

**Relation To Broader Scientific Literature:**

The key contributions of the paper are closely related to the broader scientific literature. The paper builds on existing works in parameter - efficient fine - tuning for LLMs, such as LoRA, and federated learning for LLMs. It addresses the limitations of previous approaches, like the lack of consideration for heterogeneous downstream instructions and poor understanding of the interaction between parameter - efficient fine - tuning and federated learning. By proposing FedICU, it extends the research in this area, offering a more effective way to fine-tune LLMs in federated learning.

**Theoretical Claims:**

The authors provide theoretical guarantees for the convergence of their proposed algorithms. For the Consensus - Divergence Splitting method, under standard assumptions such as smoothness, bounded gradient, and unbiased and bounded-variance gradient estimates, they prove convergence to a stationary point for non-strongly convex functions and to the unique global optimum for strongly convex functions. For the Importance - Aware Updating mechanism, they also show convergence under similar assumptions, given that the masking error is bounded. The proofs seem to be well-structured and based on established mathematical concepts in optimization theory.

---

> ### Author Rebuttal · Authors · 2025-04-01
>
> Dear Reviewer UEdC:
>
> Thank you very much for your recognition of our paper and detailed suggestions. We will answer and explain the issues in detail below.
>
> **Q1: The method relies on LoRA, which may not achieve the same performance as full parameter fine-tuning.** (Other Strengths And Weaknesses)
>
> **A1:** Thank you very much for your suggestion. Using LoRA is a common method for fine-tuning large language models today, providing significant performance improvements at relatively low cost. In the future, we will further explore how to use LoRA more efficiently to achieve the ability to fine-tune nearly all parameters.
>
> **Q2: Consider using a more flexible thresholding method in the Importance-Aware Updating mechanism.** (Questions For Authors)
>
> **A2:** Thank you very much for your suggestions. We conduct a supplementary ablation study to investigate the impact of momentum-based parameter selection and continuous-valued parameter weighting on model generalization capability, as shown in the table below. In the Momentum column, we indicate whether we include the momentum component to smooth the mask construction process. In the Smooth column, we use a smoothing mask approach based on the formula below to filter uploaded parameters and their weights. The results validate that our binary component is a simple and effective method. We will add this supplementary experiment to the appendix of our revised manuscript to enhance the description of the effectiveness of our component.
>
> *Formula: Smooth updates component.*
> $$
> W[v] =
> \\begin{cases}
> \\min{(1, \\frac{G[v]}{I[v] + G[v]})} & \\text{if } G[v] > I[v] \\\\
> 0 & \\text{other}
> \\end{cases}
> $$
>
> *Table: Experiment results of ablation study about Importance-Aware Update.*
> | Momentum |  Smooth  | MT-1 | MT-2 | MT-Final |
> | -------- | -------- | ---- | ---- | -------- |
> |    ❌    |   ✅      | 4.59 | 3.20 | 3.90     |
> |    ✅    |   ❌      | 4.60  | 3.33 | 3.97     |
> |    ✅    |   ✅      | 4.65 | 3.37 | 4.01     |
> |✅(Ours)|❌(Ours)|**4.83**|**3.43**|**4.13**|
>
> **Q3: How do you expect the performance of FedICU to scale when applied to larger models or different model architectures?** (Questions For Authors & Other Strengths And Weaknesses)
>
> **A3:** We greatly appreciate your suggestion. We've conducted supplementary experiments with Mistral-7B to verify that our method is effective across a broader range of model architectures. The results of these supplementary experiments are as follows. We will add this experiment to our revised manuscript to further support our approach.
>
> *Table: The performance of FedICU and other methods applied in the Mistral-7B. MT-1 shows the model's general capability, while MT-2 shows the model's level of contextual understanding. MT-Final serves as a comprehensive metric that combines both of the aforementioned indicators. **The results show that FedICU also demonstrates excellent performance across models with different architectures.***
> | Method    | MT-1 | MT-2 | MT-Final    |
> | --------  | ---- | ---- | ----------- |
> |  Base     | 4.10 | 3.29 | 3.69  |
> |  FedAvg   | 5.48 | 3.80 | 4.64  |
> |  Scaffold | 5.46 | 3.92 | 4.69  |
> |  FedYogi  | 5.54 | 3.85 | 4.70  |
> |  FedProx  | 5.50 | 3.83 | 4.67  |
> |  FedAdam  | 5.56 | 3.91 | 4.73  |
> |  FedAvgM  | 5.55 | 3.73 | 4.64  |
> |  Ours     | **5.58** | **4.11** | **4.84** |

---

### Official Review · Reviewer_fzFX · 2025-03-12

**Overall Recommendation:** 2

**Summary:**

This paper introduces FedICU, a framework designed to enhance fine-tuning of Large Language Models in Heterogeneous Federated Learning.

The paper presents two core innovations:
- **Consensus-Divergence Splitting**: A technique that decomposes client updates into **consensus (common capabilities)** and **divergence (domain-specific features)**. This allows the global model to **retain fundamental general knowledge** while **capturing client-specific information** effectively.
- **Importance-Aware Updating**: A method that evaluates the importance of each parameter update to **reduce unnecessary updates**, **improve communication efficiency**, and **prevent catastrophic forgetting**.

**Claims And Evidence:**

The statement in **Lines 319-322**:
   *"The significance of parameter updates varies across different downstream tasks, with some parameters showing minimal activation. These inactive parameters not only increase computational overhead during global aggregation but also can lead to catastrophic forgetting."*

   This claim is unclear.

   For example, consider a **CIFAR-10 classification scenario** where the central global model is trained to generalize across all classes:

   - **Client 1** has a well-balanced dataset covering all classes, resulting in minimal drift from the global model and smaller parameter updates.
   - **Client 2** only has samples from classes 0 and 1, causing **significant client drift** and larger parameter updates.

   In this situation, would **excluding Client 1** (which exhibits less parameter variation) actually **benefit the global model’s performance**? If so, could you clarify how inactive parameter updates contribute to catastrophic forgetting in this context?

**Essential References Not Discussed:**

In **Line 193**, the paper states:

*"Moreover, indiscriminate parameter updates risk catastrophic forgetting, where the model loses previously acquired knowledge and suffers degraded generalization capabilities."*

To strengthen this claim, it would be helpful to cite works that have empirically demonstrated **catastrophic forgetting in FL**, particularly in **image classification** settings. The following references provide relevant insights:

- **[1]** Preservation of the global knowledge by not-true self knowledge distillation in federated learning, **NeurIPS 2022**.
- **[2]** Flashback: Understanding and Mitigating Forgetting in Federated Learning, **ArXiv 2024**.
- **[3]** FedDr+: Stabilizing Dot-regression with Global Feature Distillation for Federated Learning, **TMLR 2025**.

Including these references would help contextualize the discussion of **catastrophic forgetting in FL** beyond LLM applications and provide additional experimental evidence supporting the argument.

**Experimental Designs Or Analyses:**

I did not review the experimental designs or analyses in detail but briefly looked over the results.

**Methods And Evaluation Criteria:**

1. The physical meaning of the importance measures defined in **Equations (14) and (16)** is not very intuitive. They appear to be defined arbitrarily rather than derived from a concrete motivation. It would be helpful to visualize their distributions in an **LLM FL setting** to demonstrate their necessity and relevance.

2. The justification for **Equations (17) and (18)** should be further reinforced with supporting evidence to strengthen their validity.

**Other Comments Or Suggestions:**

There are inconsistencies in the notation throughout the paper, and the lack of detailed explanations for some metrics makes it very difficult to understand.

There are several inconsistencies and ambiguities in the notation throughout the paper:

- In **Section 3**, the FL round index is denoted as **i**, but in the pseudo-code, it is written as **t ∈ [T]**. The notation should be consistent.
- In **Equation (9)**, **S_{p,q}** includes **v_p, v_q**, but it is unclear how these relate to **A_k^i** and **B_k^i**. Providing a clear explanation would improve clarity.
- In **Equation (10)**, the possible values for **j** should be explicitly stated, as the current notation is ambiguous and confusing.
- In **Equation (11)**, the temperature parameter is denoted as **T**, but in the pseudo-code, it is written as **τ (tau)**. It would be better to use a consistent notation throughout the paper.
- In **Equation (14)**, the normalization process lacks clarity regarding which samples were used to derive the **mean and variance**. Providing a precise description would enhance readability and understanding.

**Other Strengths And Weaknesses:**

Apart from the points mentioned earlier, I have no additional comments.

**Questions For Authors:**

Apart from the points mentioned earlier, I have no additional comments.

**Relation To Broader Scientific Literature:**

Unlike previous studies, this paper proposes a direct PEFT (Parameter-Efficient Fine-Tuning) method to address catastrophic forgetting, which makes it a novel contribution to the field.

**Theoretical Claims:**

I noticed that the appendix contains theorems, but since there are no theoretical claims presented in the main text, and they do not appear to be central to the paper, I did not review them.

---

> ### Author Rebuttal · Authors · 2025-03-31
>
> Dear Reviewer fzFX:
>
> Thank you for your valuable review of our paper. We hope our responses will address your concerns and improve our score.
>
> **Q1: Clarify how inactive parameter contribute to catastrophic forgetting.** (Claims And Evidence)
>
> **A1:** We apologize for the confusion in our original statement. In FedICU, we refer to inactive parameters, not inactive clients. In neural networks, only a small portion of parameters play crucial roles. To prevent clients focusing solely on local datasets and uploading all parameters that cause forgetting, FedICU calculates parameter importance and sparsely uploads parameters to balance the global model's general and domain capabilities. In the example, FedICU will upload parameters with more importance during training, balancing contributions from both clients. We conduct an experiment with results shown in the table below, demonstrating that FedICU is more stable than baseline methods and selects a substantial number of parameters from Client 1 for uploading rather than ignoring them.
>
> *Table: The accuracy of FedAvg and FedICU on CIFAR-10.*
> |Method/Round|78|79|80|81|82|83|84|
> |-|-|-|-|-|-|-|-|
> |FedAvg|41.37|42.38|42.54|41.66|42.48|40.13|40.84|
> |FedICU|45.57|46.08|46.29|42.42|46.25|46.54|46.75|
>
> *Table: The selection ratio in FedICU on CIFAR-10. The ratio means the proportion of parameters for upload. **It shows that FedICU will not ignore Client 1 for the similarity with the global model**.*
> |Client/Round|78|79|80|81|82|83|84|
> |-|-|-|-|-|-|-|-|
> |1|0.4853|0.4852|0.4866|0.4864|0.4866|0.4852|0.4855|
> |2|0.4850|0.4851|0.4849|0.4855|0.4849|0.4852|0.4857|
>
> **Q2: The physical meaning of Eq.14, 16 is not intuitive.** (Methods And Evaluation Criteria)
>
> **A2:** The internal logic of Eq. 14, 16 is that we define parameter importance based on the magnitude of updates[1, 2]. To compare client and global, we perform normalization and introduce momentum to make it stable, deriving Eq.14, 16. We add an experiment to show the distribution of importance parameters, with results shown in the table below.
>
> *Table: Parameter distribution in FL LLM. Overlap indicates important parameters globally and locally, High-I represents parameters emphasizing general ability, and High-G represents parameters emphasizing domain capability.*
> |Class/Round|5|15|25|35|45|55|65|
> |-|-|-|-|-|-|-|-|
> |Overlap|0.24|0.16|0.13|0.12|0.11|0.10|0.08|
> |High-I|0.48|0.46|0.46|0.45|0.45|0.44|0.44|
> |High-G|0.50|0.40|0.38|0.38|0.37|0.36|0.35|
>
> The results show definition's physical significance. High-I remains stable while decreasing High-G proportions indicate better identification of core domain parameters. Declining overlap suggests clearer functional partitioning, effectively separating general and domain-specific knowledge.
>
> **Q3: The justification for Eq.17, 18 should be reinforced to strengthen validity.** (Methods And Evaluation Criteria)
>
> **A3:** Sparsely updating model is an approach to mitigate catastrophic forgetting [2, 3]. In FedICU, We construct masks and sparsely update models (Eq.17, 18) to mitigate forgetting. To further demonstrate the effectiveness, we conduct an experiment as shown in the table below. For Momentum, we test whether to incorporate momentum components, and for Smooth, whether to use binary or smooth updates based on the formula below. Results show our method achieves the best performance, demonstrating its effectiveness.
>
> *Formula: Smooth updates component.*
> $$
> W[v] =
> \\begin{cases}
> \\min{(1, \frac{G[v]}{I[v] + G[v]})} & \\text{if } G[v] > I[v] \\\\
> 0 & \\text{other}
> \\end{cases}
> $$
>
> *Table: Experiment results of ablation study. MT-1 shows the model's general capability, while MT-2 shows the model's level of contextual understanding. MT-Final is the metric combining both of the two indicators.*
> |Momentum|Smooth|MT-1|MT-2|MT-Final|
> |-|-|-|-|-|
> |❌|✅|4.59|3.20|3.90|
> |✅|❌|4.60|3.33|3.97|
> |✅|✅|4.65|3.37|4.01|
> |✅(Ours)|❌(Ours)|**4.83**|**3.43**|**4.13**|
>
> [1]Learning both weights and connections for efficient neural network.**NeurIPS, 2015**.
> [2]A simple and effective pruning approach for large language models.**arXiv 2023**.
> [3]Finding sparse, trainable neural networks.**arXiv 2018**.
>
> **Q4: Some essential references are not discussed.** (Essential References Not Discussed)
>
> **A4:** Thank you for suggesting additional references. We will incorporate these into the revised manuscript.
>
> **Q5: Some inconsistencies in the notation.** (Other Comments Or Suggestions)
>
> **A5:** Thank you for pointing out the unclear expressions.
> - For the second, the vectors are defined in Lines 252-261. Specifically, we perform vector-wise aggregation by processing $A_k^i$ column-by-column to obtain $r$-dimensional vectors $v$, while for $B_k^i$, we process it row-by-row.
> - For the third, $j \in \{1, 2, ..., n\}$.
> - For the fifth, we use model gradients for mean and variance.
>
> We will correct all of them in the revised manuscript.

---

### Decision · Program_Chairs · 2025-05-01

**Decision:**

Accept (poster)

**Comment:**

The reviewers highlighted this paper's novelty, thorough analyses, and convincing experiments. The authors also provided reasonable answers to the doubts about the details of the proposed dataset and method, the authors further explain the proposed method and add more experiments to support it. Moreover, FedICU performs well across different model architectures, demonstrating its generalizability. The reviewers’ concerns were well-addressed. Therefore, the decision is to recommend Acceptance.